# Toxicity, Morality, and Speech Act Guided Stance Detection

**Apoorva Upadhyaya, Marco Fisichella, Wolfgang Nejdl**
L3S Research Center, Leibniz Universität Hannover, Germany
{upadhyaya, mfisichella, nejdl}@l3s.de

## Abstract

In this work, we focus on the task of determining the public attitude toward various social issues discussed on social media platforms. Platforms such as Twitter, however, are often used to spread misinformation, fake news through polarizing views. Existing literature suggests that higher levels of toxicity prevalent in Twitter conversations often spread negativity and delay addressing issues. Further, the embedded moral values and speech acts specifying the intention of the tweet correlate with public opinions expressed on various topics. However, previous works, which mainly focus on stance detection, either ignore the speech act, toxic, and moral features of these tweets that can collectively help capture public opinion or lack an efficient architecture that can detect the attitudes across targets. Therefore, in our work, we focus on the main task of stance detection by exploiting the toxicity, morality, and speech act as auxiliary tasks. We propose a multitasking model TWISTED that initially extracts the valence, arousal, and dominance aspects hidden in the tweets and injects the emotional sense into the embedded text followed by an efficient attention framework to correctly detect the tweet's stance by using the shared features of toxicity, morality, and speech acts present in the tweet. Extensive experiments conducted on 4 benchmark stance detection datasets (SemEval-2016, P-Stance, COVID19-Stance, and ClimateChange) comprising different domains demonstrate the effectiveness and generalizability of our approach.

## 1 Introduction

Social media platforms are often used to express public opinions and raise awareness about various issues and current problems in society (Lineman et al., 2015; Kaur and Chahal, 2018; Li et al., 2021a). However, conversations on platforms like Twitter can lead to users being exposed to false, misinformed messages, often resulting in polarizing beliefs and echo chambers (Garimella and Weber, 2017; Li et al., 2021a). A recent American Press article[1] claims that misinformation about climate is widespread on Twitter and often delays appropriate climate action. In addition, a study conducted by MIT researchers[2] showed that false news spread 6 times faster on Twitter than truthful news. Therefore, it is necessary to recognize the attitudes of the posts so as to curb the spread of misinformation. In our work, we perform stance detection of tweets to detect the opinion of the post, whether it is in favor, against, or none toward a target topic. There is a wealth of research on identifying attitudes in social media. Several popular and benchmark stance datasets exist in the literature such as SemEval-2016 (Mohammad et al., 2016), P-Stance (Li et al., 2021b), COVID-19 (Glandt et al., 2021), and ClimateChange (Upadhyaya et al., 2023b) that focus on various social issues such as feminism, abortion, COVID19, climate, and many current crises whose public opinions need to be identified in order to address these issues and change society for the better. We also use these 4 benchmark datasets in our study, which covers different domains, to get an assessment of generalizability of our approach.

Existing literature suggests that the tweet content consists of abusive and insulting statements, leading to the spread of hatred, bullying, and negativity in the public sphere, often disturbing social peace (Hosseinmardi et al., 2015; Cheng et al., 2017; Pavlopoulos et al., 2019). Moreover, the moral values in the tweets correlate with the different public opinions, which helps to understand the intentions behind each expressed attitude, which in turn helps to culturally integrate society and mitigate such differences (Rezapour et al., 2021). In the popular benchmark datasets, we also note

---

[1]https://apnews.com/article/elon-musk-twitter-inc-technology-science-social-media-a7e2e3214abb4470dcb6e2837aa39c2e

[2]https://news.mit.edu/2018/study-twitter-false-news-travels-faster-true-stories-0308

that the speech act of the tweets, i.e., the form of expression, statement, question, or thought, also differs significantly across viewpoints on different topics (refer Appendix B.2), which helps to determine the communicative intent of the viewpoint, further supporting the stance task. The following are few examples of tweets from public datasets: *(ii)Feminism (against):* "@HiddenTara No one wants to see Feminists naked, so the petty, vindictive bit**** want to drag everyone else down to their level *(toxic,harm,expression)*"; *(ii)Biden (favor):* "Lets Go Joe #TeamJoe #Go-Joe #Biden2020 *(non_toxic, care, loyalty, suggestion)*". These relationships between toxic, moral, and speech act labels associated with the tweets' stances led us to use them to decipher their underlying attitudes.

While several works (Vychegzhanin and Kotelnikov, 2021; Wang et al., 2020; Li and Caragea, 2019) focused on the linguistic properties of the input text for stance, they often lack an advanced architecture in the form of a better encoder trained exclusively on tweets, thus failing to extract the masked toxic and moral aspects from tweets in a multi-tasking form. Other recent works such as MEMOCLiC (Upadhyaya et al., 2023b), SP-AMT (Upadhyaya et al., 2023c) have used sentiment, emotion, and toxicity aspects for stance detection, but these models focus primarily on climate change and often suffer from the drawbacks of detecting overall stance in the case of composite tweets with contrasting emotions or complexity of tweet content due to sarcasm. It has been proven that the valence, arousal, and dominance features help in analyzing emotional tone and its intensity (Osgood et al., 1957; Russell, 1980, 2003; Mohammad, 2022), which eventually supports the detection of hidden intent or any sarcasm present in the tweet (Vitman et al., 2022). Therefore, our approach leverages these interdependencies to extract additional features of valence, arousal, and dominance (VAD) along with insult, morality, and speech act to understand the psychology of the tweet and gain emotional insights. These features can then ultimately help identify the collective meaning and intent of the tweet, aiding in the detection of attitudes and thus addressing the drawbacks of the presence of sarcasm and implicit stance from previous works.

The main contributions of our work are as follows: *(i.)* To the best of our knowledge, this is the first cross-sectional study to incorporate toxicity, morality, and speech act to determine the tweet's stance. This opens a new dimension in psychological and social science research to investigate in-depth the interdependencies between these toxic, moral, speech acts and public opinion. *(ii.)* We propose a multi-tasking system TWISTED that performs the main task of stance detection (SD) by using toxicity detection (TD), morality classification (MC), and speech act classification (AC) as auxiliary tasks. Our TWISTED model first extracts the VAD features to capture the associated emotions and intentions of the tweet. EmoSenseInjector then induces the emotional sense to the tweet by integrating the embedded text with the VAD vectors. The emotionally enriched tweet is followed by an efficient attention framework that uses the task-specific and shared toxic, moral, and act features to identify the attitude of the tweet (favor/against/none). *(iii.)* Extensive experiments are conducted on 4 benchmark stance detection datasets (SemEval-2016, P-Stance, ClimateChange, COVID-19-Stance) and the reported experimental results demonstrate the usefulness of our approach. The code and datasets with annotations are available here[3].

## 2 Related Works

Stance detection has been investigated in a number of studies. Some of the existing works focus on exploring the impact and influence of polarizing attitudes (Chitra and Musco, 2020), while others use the characteristics of tweet texts or networks to classify attitudes towards a target domain by developing machine and deep learning algorithms (Dutta et al., 2022; Upadhyaya et al., 2023d). Since it is critical to understand public attitudes toward any pressing issue, our study also focuses on detecting the stance of a tweet text toward an urgent social issue. This can help technology companies and government agencies monitor public opinions and intervene to curb the spread of fake news, misinformation, or online hatred that can harm the peace.

Previous works have enriched the stance task with popular datasets covering controversial topics such as feminism, abortion, political leaders, COVID-19, and the climate crisis (Mohammad et al., 2016; Glandt et al., 2021). Various existing studies have primarily focused on the SemEval

---

[3]https://osf.io/en4rd/?view_only=86e383c5bf1441d0a19f7a968ddbd6e6

2016 dataset (Vychegzhanin and Kotelnikov, 2021; Wang and Wang, 2021) while others have focused on political and COVID domains (Zheng et al., 2022; He et al., 2022). The climate change domain, which focuses on the climate crisis, has also been recently explored for stance task (Upadhyaya et al., 2023a,b). However, in our study, we use these 4 benchmark datasets and investigate the performance of stance task that span a variety of contrasting domains with different embedding spaces of toxic, moral, and act features to understand the practicality of our approach.

Recently, research has explored multitasking and federated learning for various classification tasks such as hate speech, toxicity detection, and infrastructure damage identification, as the MTL paradigm aims to improve generalization performance by learning multiple related tasks at once (Priya et al., 2020; Vaidya et al., 2020; Mishra et al., 2021; Badar et al., 2023; Younis et al., 2023). However, as far as we know, this is the first cross-sectional study to utilize toxicity, morality, and speech act detection tasks in a multitask learning framework to identify the attitude of a tweet.

In addition, the existing literature pointing to the correlation between toxic and moral aspects in relation to public opinions expressed on Twitter motivated us to use these features as auxiliary tasks (Rezapour et al., 2021; Upadhyaya et al., 2023b). Different forms of speech act further help to decode the pragmatics of the author of the tweet and to identify the attitude encoded in the tweet (Saha et al., 2021). Models such as ESD (Vychegzhanin and Kotelnikov, 2021) and HAN (Wang et al., 2020) exploit the linguistic characteristics of tweets by incorporating sentiments, stylistic, and stance-indicative features. However, these earlier studies did not use the better embedding techniques by using the BERTweet encoder along with extracting VAD features that provide the emotional tone and intensity that better identify different attitudes even when similar emotions or feelings are present. Recent climate stance models use emotion and toxicity to identify stance (Upadhyaya et al., 2023b) but suffer from problems with the presence of contrasting emotions or sarcasm in the stances. However, our model addresses these problems by extracting the VAD features along with the toxic and moral aspects, which together understand the overall intent of expressed public opinion and aid the stance task.

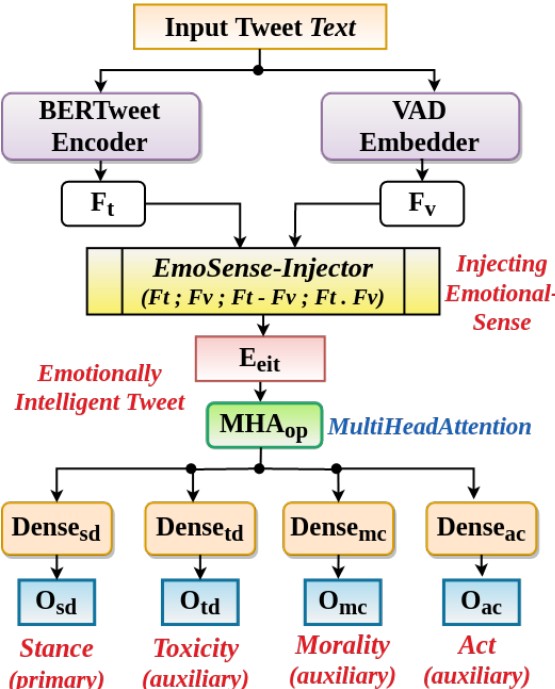

Figure 1: Architectural overview of our proposed TWISTED approach

## 3 Methodology

**Problem Definition** Given a tweet, we propose a stance detection approach that combines the tweet text with the extracted valence, arousal, and dominance features while taking advantage of toxic, moral, and speech act aspects associated with the tweet to efficiently detect the stance toward a target (favor/against/none).

Our proposed framework TWISTED comprises of the model components: *BERTweet-Encoder, VAD-Embedder, EmoSense-Injector, MultiHeadAttention, and Output Layer*. Figure 1 represents the overview of our TWISTED approach. The tweet is passed in parallel to the BERTweet Encoder to generate the embedding of the tweet text, and to the VAD Embedder to extract the valence, arousal, and dominance features related to the tweet. The encoded text and the VAD features are passed through the EmoSense-Injector which is responsible for integrating the emotional sense in the tweet and obtaining the emotionally intelligent representation of the input tweet. This emotion-aware tweet is then followed by the MultiHeadAttention layer to focus on the most relevant input features, followed by the task-specific dense and output layers to finally obtain the outputs of the main task of SD and the secondary tasks of TD, MC, and AC.

## 3.1 BERTweet Encoder

The input tweet text is converted into a sequence of vectors using the BERTweet encoder (Nguyen et al., 2020), which is specially trained on tweets to embed each word in the tweet text. First, the tweets are tokenized and divided into sequences of tokens of the form [CLS] t1,t2..tn, where [CLS] is a special token that marks the beginning of a tweet. In each minibatch, the input tokens are padded to a maximum sequence length (say m). The final state corresponding to the first token [CLS] is used as the overall representation of a tweet. BERTweet Encoder generates embeddings of size 768 dimensions ($d_t$) for each input token, which is then flattened, resulting in $F_t \in R^{m(d_t)}$.

## 3.2 VAD Embedder

We employ NRC-VAD Lexicon (Mohammad, 2018) to acquire the valence, arousal, and dominance (VAD) features of the input tweet. The preprocessed input text (the preprocessing steps are described in the Appendix B.1) consists of tokens generated by the NLTK TweetTokenizer and then passed to the NRC-VAD lexicon. The lexicon consists of 20,000 English words, with each word having a probabilistic score for all three aspects of valence, arousal, and dominance. The VAD scores of each token in the input text are then multiplied by the occurrence of that token for all 3 dimensions (V, A, and D), summed, and aggregated to create a single vector of dimension 3 ($d_v$) for each input tweet. The final feature representing the VAD embedding space for each input tweet is thus $F_v \in R^{d_v}$.

## 3.3 EmoSense-Injector

The component is accountable for incorporating the emotional sense and intensity extracted from the VAD features into the embedded tweet text to obtain an overall representation of the input tweet in accordance with the associated emotions. The text ($F_t$) and VAD ($F_v$) feature vectors are initially passed through a fully connected layer of dimension $d_e$. The resulting text and VAD vectors are then fused using absolute difference and element-wise product (refer Equation 1), as the fusion technique has been proven efficient in various classification studies (Mou et al., 2015). The integrated output is then reshaped ($E_{eit} \in R^{1 \times 4(d_e)}$) and finally represents the emotionally intelligent tweet that captures the essence of the emotions present along with the semantic and syntactic contextual text representation.

$$E_{eit} = [F_t; F_v; F_t - F_v; F_t \odot F_v] \qquad (1)$$

## 3.4 MultiHeadAttention

We use the MultiHeadAttention similar to (Vaswani et al., 2017) based on the concept of query (Q), key (K), and value (V) to jointly capture the most relevant parts of the input feature from different representation subspaces. Instead of performing a single attention layer (refer Equation 3), Q, K, and V are linearly projected h times (where h denotes the number of heads) and Attention function (Equation 3) is performed in these different learned projections of Q, K, and V in parallel. These outputs of parallel attention functions are concatenated and once again projected, resulting in the final attention vector. To perform MHA, we employ tfa.layers.MultiHeadAttention layer[4], where the output of EmoSense-Injector ($E_{eit}$) is fed as query, key, and value to the MultiHeadAttention layer of TensorFlow.

$$Q = E_{eit}, K = E_{eit}, V = E_{eit} \qquad (2)$$
$$Attention = softmax(QK^T)V \qquad (3)$$
$$mha = MultiHeadAttention(h_s, h) \qquad (4)$$
$$MHA_{op} = mha([Q, K, V]) \qquad (5)$$

Equations 4 and 5 represent the MultiHeadAttention TensorFlow layer being implemented where $h$ denotes the number of heads indicating the number of times the single Attention function (shown in Equation 3) is performed, $h_s$ specify the size of the head, and $E_{eit}$ is used as query, key, and value, resulting in $MHA_{op} \in R^{1 \times 4(d_e)}$.

## 3.5 Output Layer

The shared output of MultiHeadAttention ($MHA_{op}$) component is passed through the task-specific dense layers of dimension [$d_d$] ($Dense_{sd}$, $Dense_{td}$, $Dense_{mc}$, $Dense_{ac}$) followed by four softmax layers for capturing the final output specific to each task of SD ($O_{sd}$), TD ($O_{td}$), MC ($O_{mc}$), and AC ($O_{ac}$) separately. The integrated loss function (L) of our TWISTED framework is realized in equation 6 indicating loss for each specific task of SD ($L_{sd}$), TD ($L_{td}$), MC ($L_{mc}$), and AC ($L_{ac}$):

$$L = p * L_{sd} + q * L_{td} + r * L_{mc} + s * L_{ac} \quad (6)$$

---

[4]https://www.tensorflow.org/addons/api_docs/python/tfa/layers/MultiHeadAttention

where p, q, r, and s are the values between 0 and 1 and show the ratio of loss of each task to total loss.

# 4 Experimental Set-up

## 4.1 Datasets

We use the 4 benchmark datasets of stance detection containing different targets: *(i) SemEval-2016 (Mohammad et al., 2016):* is a popular dataset used in SemEval-2016 shared task 6.A that covers Atheism, Climate Change is a Real Concern, Feminist Movement, Hillary Clinton, and Abortion as targets with tweets having favor, against, or neutral stances. *(ii) P-Stance (Li et al., 2021b):* is a political stance dataset containing $7,953$ annotated tweets for "Donald Trump", $7,296$ for "Joe Biden" and 6,325 for "Bernie Sanders" as target domains with favor and against stances. *(ii) ClimateChange (Upadhyaya et al., 2023b):* In this benchmark dataset, $8,881$ climate tweets are included that have believe, deny, and ambiguous views regarding climate change. *(iv) COVID-19-Stance (Glandt et al., 2021)* is a dataset of pandemic-related $6,133$ tweets on four controversial topics ("Anthony S. Fauci, M.D (Director of the National Institute of Allergy and Infectious Diseases)," "stay at home orders" "wearing a face mask," and "keeping schools closed") and consists of stances ("FAVOR", "AGAINST", "NONE") on these topics.

### 4.1.1 Data Annotation

All publicly available datasets consist of annotations of stances, however, to find the toxic, moral, and speech act labels, we follow the below weak-supervised approaches.

*Toxicity Detection (TD)* Similar to previous works (Hickey et al., 2023; Upadhyaya et al., 2023b), the Perspective API developed by Jigsaw and Google's Counter Abuse Technology team in Conversation-AI (Hosseini et al., 2017) is used to provide labels for the TD task. We use the probability value (0-1) for the "toxicity" metric returned by the API. After careful analysis of various thresholds and in-line with existing literature (Upadhyaya et al., 2023b), we decided that if the value of the "toxicity" attribute is $\geq 0.5$, we consider the tweet to be toxic so as not to miss any toxic content, otherwise, the tweet will be labeled as *non_toxic*.

*Morality Classification (MC)* We leverage the extended version of the Moral Foundation Dictionary (eMFD) (Hopp et al., 2021) for MC task annotation. The open-source Python library "eMFD-

score"[5] provides the scores for the 10 moral classes according to the sociological theory of Moral Foundations Theory (MFT), which identifies morality as a set of vice virtues, including care–harm, fairness–cheating, loyalty–betrayal, authority–subversion, and purity–degradation (Graham et al., 2013). The tweet text is fed into the eMFDScore library (with DICT_TYPE ='emfd', PROB_ MAP = 'all', SCORE_ METHOD = 'bow', OUT_ METRICS = 'vice-virtue' as parameter values), which first preprocesses the text and assigns a list of 10 foundation scores indicating the degree to which each tweet reflects the moral values. To facilitate the performance of morality as a multi-label classification task, in our study, after manual review and careful analysis, we convert the list of probabilistic scores into a list of 0s and 1s indicating the presence/absence of the corresponding moral attribute. To achieve this, we first compute the mean ($mean$) and standard deviation ($\sigma$) of the list of 10 moral scores and consider $threshold$ as ($mean + \sigma$). If the value of an attribute is $> threshold$, we marked the presence of the corresponding attribute as "1", otherwise as "0". The final label of each tweet for the MC task results in the list of 10 elements being 0 or 1 (for example: MC label=[1,0,1,0,0,0,0,0,1,0], suggesting the presence of the moral values care, fairness, and purity/sanctity).

*Speech Act Classification (SC)* We use the pre-trained benchmark speech act classifier trained on the gold-standard speech act tweet corpus (Saha et al., 2021), which contains 6749 tweets, to provide weak speech act labels for the instances of the 4 stance datasets used in our study. Since the pre-trained classifier performs better on the 5-class classification of speech acts, we also extract the speech act labels for the tweets of the 4 stance datasets in the form of expression, question, statement, suggestion, and others.

*The data pre-processing, % distribution of auxiliary features, and manual verification of labels generated by weakly supervised approaches are mentioned in Appendix B.*

## 4.2 Implementation Details

**Hyperparameters:** maximum sequence length ($m$): 128; BERTweet embedding dimension ($d_t$): 768; VAD feature dimension ($d_v$): 3; EmoSense-

---

[5] https://github.com/medianeuroscience/emfdscore

Injector dense layer dimension ($d_e$) [with ReLu activation]: 128; MultiHeadAttention parameters: 8 [number of heads ($h$)], 64 [head_size ($h_s$)]; task-specific dense layer dimension ($d_d$) [with ReLu activation]: 128, output neurons/channels for *auxiliary tasks*: 2 [softmax activation] (TD), 10 [sigmoid activation] (MC), and 5 [softmax activation] (AC), loss function for *auxiliary*: categorical cross-entropy for TD ($L_{td}$) and AC ($L_{ac}$); binary cross-entropy for MC ($L_{mc}$), output neurons for *main SD task*: *P-Stance dataset*= 2 output neurons and binary cross-entropy for SD (presence of "favor" and "against" stances); *SemEval, ClimateChange, COVID datasets*= 3 output channels and categorical cross-entropy loss for SD ($L_{sd}$), Optimizer: Adam (0.0001 learning rate), batch_size: 32. The best parameter values are selected using TPE in the Python library Hyperopt[6], which minimizes loss functions. The loss weights for all tasks are fine-tuned using Grid Search from Scikit-learn (SD (*p*)=1, TD (*q*)=0.5, MC (*r*)=0.3, and AC (*s*)=0.4).

**Evaluation Metrics:** We use already available train, validation, and test splits of the SemEval-2016, P-Stance, and COVID-19-Stance datasets, perform 5 independent runs of our framework to account for variability, and report the macro-average of the F1-score as the evaluation metric. For SemEVAL-2016 (does not consider None stance label for performance evaluation) and P-Stance (consists of 2 stance labels), $F_{avg} = (F_{favor} + F_{against})/2$, similar to their proposed works (Mohammad et al., 2016; Li et al., 2021b) and $F_{avg} = (F_{favor} + F_{against} + F_{none})/3$ for COVID-19-Stance dataset (Glandt et al., 2021; He et al., 2022). In-line with existing work that proposed ClimateChange data (Upadhyaya et al., 2023b), we implement our TWISTED approach with 5-fold cross-validation technique and report the average and standard deviation of macro-variant of precision, recall, F1-score as the performance metrics.

**Environment Details**: described in Appendix C.

### 4.3 Baselines

All experimental results of the baselines used on all the datasets are retrieved from the original papers.

**SemEval-2016** We compare TWISTED with the recent works that use this dataset: Semi-Supervised(Model3) (Reveilhac and Schneider, 2023), MEMOCLiC (Upadhyaya et al., 2023b), SP-AMT (Upadhyaya et al., 2023c), MT-LRM-BERT

---

| Model | Atheism $F_{avg}$ | Climate $F_{avg}$ | Feminism $F_{avg}$ | Hillary $F_{avg}$ | Abortion $F_{avg}$ | Mac $F_{avg}$ |
|---|---|---|---|---|---|---|
| TWISTED (Ours) | 79.57* | 78.74* | **76.09** | **78.29** | **76.67** | **77.87** |
| MEMOCLiC | 74.39 | 64.51 | 63.62 | 75.84* | 71.36 | 69.94 |
| Model3 | **83.00** | 70.00 | 63.00 | 67.00 | 70.00 | 70.6 |
| MT-LRM -BERT | 76.14 | 53.05 | 63.12 | 74.67 | 70.32 | 67.46 |
| SP-MT | 69.5 | 63.5 | 63.2 | 67.5 | 70.5 | 66.84 |
| WKNN | 73.52 | **79.95** | 72.99* | 75.02 | 75.74* | 75.44* |
| S-MDMT | 69.50 | 52.49 | 63.78 | 67.20 | 67.19 | 64.03 |
| ESD | 66.64 | 43.82 | 62.85 | 67.79 | 64.94 | 61.20 |
| HAN | 70.53 | 49.56 | 57.50 | 61.23 | 66.16 | 61.00 |
| AT-JSS -LEX | 69.22 | 59.18 | 61.49 | 68.33 | 68.41 | 65.33 |
| SVM-ngram | 65.19 | 42.35 | 57.46 | 58.63 | 66.42 | 58.01 |

Table 1: Results of Stance task on SemEval-2016 with Baselines [* denotes the 2nd-best performer]

(Fu et al., 2022), MDMT (Wang and Wang, 2021), ESD (Vychegzhanin and Kotelnikov, 2021), HAN (Wang et al., 2020), AT-JSS-LEX(Li and Caragea, 2019), and SVM-ngram (Sobhani et al., 2016).

**P-Stance** We take the baseline and best-performing models reported in recent works (Liu et al., 2023; Zheng et al., 2022; He et al., 2022): In-Domain In-target variant RoBERTa-base (Liu et al., 2023) (we re-run the model with respect to "Trump", "Biden", and "Bernie" separately as results reported in the paper are on the complete dataset), WS-BERT-Dual (He et al., 2022), BERTweet (Nguyen et al., 2020), BERT (Devlin et al., 2019), PGNN (Huang and Carley, 2018), TAN (Du et al., 2017), and BiCE (Augenstein et al., 2016).

**ClimateChange** We use the baselines as is from the work that proposed data (Upadhyaya et al., 2023b) and report average of macro-variant of precision, recall, and F1-score: SP-AMT (Upadhyaya et al., 2023c), RoBERTa-Base (Vaid et al., 2022), MT-LRM BERT (Fu et al., 2022), S-MDMT (Wang and Wang, 2021), ESD (Vychegzhanin and Kotelnikov, 2021), HAN (Wang et al., 2020), MNB (Kabaghe and Qin, 2020), and DNN (Chen et al., 2019).

**COVID-19-Stance** Below are best-performing and baseline models mentioned in recent literature (Upadhyaya et al., 2023d; Zheng et al., 2022; He et al., 2022): STASY (Upadhyaya et al., 2023d), WS-BERT-Dual (He et al., 2022), CT-BERT-DAN (Glandt et al., 2021), GCAE (Xue and Li, 2018), ATGRU (Zhou et al., 2017), and TAN (Du et al., 2017).

### 5 Results

*Please note that we report the results of the main task SD as we aim to improve the performance of the primary task by using other auxiliary tasks in our current study.*

**SemEval-2016** Table 1 shows that our TWISTED

| Model | Trump | Biden | Sanders | Avg. |
|---|---|---|---|---|
| TWISTED | **87.3** | **89.7** | **84.6** | **87.2** |
| RoBERTa-Base | 82.6 | 81.3 | 78.1 | 80.7 |
| WS-BERT-Dual | 85.8 | 83.5 | 79.0 | 82.8 |
| BERTweet | 85.2 | 82.5 | 78.5 | 82.1 |
| BERT | 78.3 | 78.7 | 72.5 | 76.5 |
| PGCNN | 76.9 | 76.6 | 72.1 | 75.2 |
| TAN | 77.1 | 77.6 | 71.6 | 75.1 |
| BiCE | 77.2 | 77.7 | 71.2 | 75.4 |

Table 2: Results for Stance Detection on P-Stance

| Model | Precision | Recall | F1 score |
|---|---|---|---|
| - | Avg/St.dev | Avg/St.dev | Avg/St.dev |
| TWISTED (Ours) | **95.19/0.46** | **96.11/0.23** | **96.05/0.32** |
| MEMOCLiC | 92.06/0.81 | 95.44/0.29 | 93.76/0.62 |
| RoBERTa-Base | 83.38/1.55 | 85.24/1.28 | 84.69/1.89 |
| SP-AMT | 87.95/1.11 | 90.01/1.80 | 89.29/1.31 |
| MT-LRM-BERT | 87.12/1.61 | 88.70/0.99 | 88.59/1.29 |
| S-MDMT | 86.12/1.02 | 88.67/0.39 | 86.91/0.44 |
| ESD | 81.55/1.72 | 84.39/2.05 | 83.28/2.31 |
| HAN | 84.61/1.22 | 84.23/1.78 | 84.54/1.65 |
| MNB | 78.11/0.66 | 79.51/0.73 | 78.43/1.33 |
| DNN | 77.64/1.58 | 76.38/1.08 | 77.15/1.18 |

Table 3: Results for Stance Detection on ClimateChange

model outperforms the other baselines with a overall $Mac\text{-}F_{avg}$ of 77.87, resulting in an average F1 score performance improvement of 3.22% to 34.24%. This demonstrates the importance of using features such as toxicity, morality, and speech act in the context of the SD task. In addition, we find that TWISTED performs better, especially in the feminism, Hillary, and abortion domains, because of a clearer separation between favor, against, and none stances in terms of their hidden offensive and moral aspects (see Table 6 in Appendix). However, our approach does not beat the Model3 baseline on "atheism" domain and performs second best, although the tasks TD and AC contributed well due to the clear distinction in their embedding spaces with respect to the different attitudes, but the moral values between the 3 stance classes are much more closely aligned, making it difficult for TWISTED to distinguish between all stance classes (refer Appendix Table 6). WKNN achieves a slightly better result than TWISTED (78.74) with 79.95 $F_{avg}$ on "climate". This is due to the fact that the dataset contains only 29 tweets from deniers, which makes it difficult to capture different speech act and moral labels such as betrayal, subversion, purity, and question (Appendix Table 6) due to the very low availability of token words, however, the significant differences in VAD features (see VAD features in the dataset[3]) between stance labels and the clearer separation between toxic and non-toxic content embedded using the BERTweet model trained exclusively on tweets contribute to TWISTED being the second best performer among baselines on "climate" target domain.

**P-Stance** From Table 2, our TWISTED model performs better than Trump in Sanders and Biden, as shown by the 1.7%, 7.1%, and 7.4% improvement in F1 results, respectively, compared to the second-best performing baseline (WS-BERT-Dual). This is mainly due to the fact that supporters and opponents in Sanders (28.98 diff.) and Biden (26.14

diff.) domains show a better-dividing line in terms of toxic characteristics than Trump (17.23 diff.) (refer Appendix Table 7). Nevertheless, TWISTED, with an average macro-F1 score of 87.2 and an overall average improvement of 11.61%, outperforms the baselines, justifying the usefulness of our multitasking approach in inducing emotional meaning in tweets using VAD feature extraction and an efficient attention framework in the political domain.

**ClimateChange** Table 3 indicates that TWISTED performs better than the other baselines with an average percentage improvement of 13.14% in the F1 score. Although MEMOCLiC extracts the emotional and toxic features in a multitasking environment to identify the underlying attitude of the tweet, but the performance power of our EmoSense injector, which injects the VAD features, understanding not only the emotional quotient of the tweet, but also the intensity and tone along with the hidden moral and linguistic features, is one of the main reasons contributing to the success of TWISTED compared to the recent works MEMOCLiC, SP-AMT. We note that deniers' tweets have a high proportion of bad moral values (harm, betrayal, and humiliation) and toxic content (Appendix Table 8), demonstrating the usefulness of our approach for government agencies or technology companies that need to monitor such content before it leads to a delay in appropriate climate action. Models such as MEMOCLiC, SP-AMT, and MT-LRM-BERT, which take advantage of toxicity, emotion, sentiment, or opinion formation tasks, perform better than other baselines, proving the effectiveness of these auxiliary tasks for the SD.

**COVID-19-Stance** Table 4 proves the efficacy of our TWISTED model in detecting attitudes that occurred during the controversial COVID topics

| Model | Fauci | Home | Mask | Schools | Avg. |
|---|---|---|---|---|---|
| TWISTED | **91.05** | **91.63** | 85.29 | **89.78** | **89.44** |
| STASY | 88.04 | 89.19 | 86.15 | 85.71 | 87.27 |
| WS-BERT-Dual | 83.6 | 85 | **86.6** | 82.2 | 84.4 |
| CT-BERT-DAN | 83.2 | 78.7 | 82.5 | 71.7 | 79 |
| GCAE | 64 | 64.5 | 63.3 | 49 | 60 |
| ATGRU | 61.2 | 52.1 | 59.9 | 52.7 | 56.5 |
| TAN | 54.7 | 53.6 | 54.6 | 53.4 | 54.1 |

Table 4: Results for Stance Detection on COVID-19

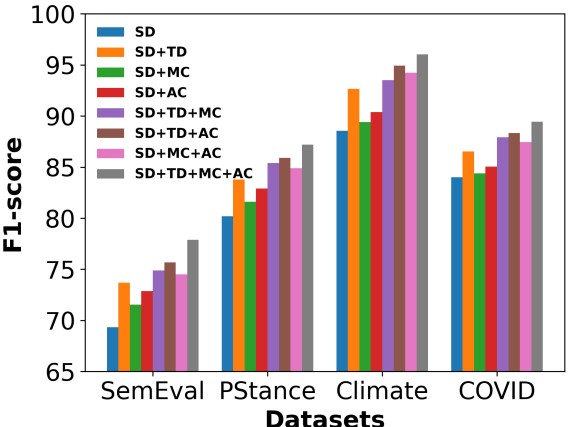

Figure 2: Ablation Study: Performance of 4 datasets with different combinations of auxiliary tasks for the main task of stance detection.

"Fauci", "Home", and "Schools" with the exception of the topic "wearing a face mask (Mask)". We note that WS-BERT-DUAL performs better on the "MASK "topic than TWISTED, as the data distribution of the aspects of "toxic", "non-toxic", and "expression of speech act" are close among supporters and opponents, and the percentage of vices in the form of "harm", "cheating," and "degradation" are similar among the against and none stance labels (see Appendix Table 9), which prevents the TWISTED approach from clearly distinguishing the polarized labels for the SD task. It can also be seen from Table 4 that the STASY model, which uses the sentiments and temporal aspects present in the tweets belonging to different public opinions, gives better results than other baselines, proving the need to identify hidden aspects in the tweets that can be used for the task SD. However, the average macro F1 score of 89.44 achieved by TWISTED suggests that our approach is also well suited for the important social issues in the domain COVID.

Compared with the baselines, the results of TWISTED were statistically significant (under t-tests ($p < 0.05$). The better performance of TWISTED in various target areas such as feminism, abortion, politics, climate, and COVID and comparable performance in other subject domains proves the usefulness of our approach and the generalizability of our system.

**Ablation Study** Here we examine the effects of single-task, all the auxiliary tasks and their combinations for the stance task as part of the ablation study. Figure 2 shows the macro-average F1 score of all 4 datasets when performing the main task SD with different combinations of auxiliary tasks such as SD (Single-task), SD+TD, SD+MC, SD+AC, SD+TD+MC, SD+TC+AC, SD+AC+MC, SD+TD+MC+AC (all). Figure 2 clearly shows that the single-task variant of our TWISTED approach is the worst performer when compared with other auxiliary task combinations, suggesting the potency of using auxiliary tasks for SD. It is fur-

ther evident from the figure, toxic and non-toxic features associated with tweet text help the attitude task more effectively than the other tasks MC and AC due to their significant differences in distribution among "favor", "against", and "none" stances (see Appendix Tables 6, 7, 8, and 9). This also justifies the assignment of higher loss weight to TD task than MC and AC tasks. Moreover, the 3-task combination SD+TD+AC contributes more than the tasks SD+TD+MC and SD+MC+AC, further proving that expression, statement, query, and other speech acts are more consistent with the toxic aspects of tweets. However, the performance improvement of TWISTED on all 4 data sets when all 3 secondary tasks are used (SD+TD+MC+AC) confirms the importance of all 3 tasks for main SD.

The **case studies** and possible **error scenarios** of our framework are described in Appendix A and Table 5. Furthermore, the ablation study showing the importance of our proposed components for the ClimateChange and COVID-19 datasets is mentioned in the Appendix D.

## 6   Conclusion

In this work, we present the first cross-sectional study focusing on the problem of stance detection using the associated toxic, moral, and speech act traits present in tweets. Our proposed approach is enriched with the emotional quotient of the tweet by extracting the VAD features and further integrating them into the embedded tweet, thus evoking the emotional intent of the tweet. The experimental results on 4 benchmark datasets indicate that multi-tasking enhances the performance of the stance

task by exploiting the efficient attention and encoder frameworks along with the auxiliary tasks. This demonstrates that the model is widely applicable in multiple domains, proving the generalizability of our proposed approach. In the future, we plan to include additional tasks such as entity-recognition and aspect-based sentiment detection, which could help disambiguate the attitudes and conflicting emotions in a tweet to predict a more accurate classification of polarized attitudes toward a target domain.

## Limitations

The error scenarios listed in Appendix A.1 (Table 5), which arise from the proximity of auxiliary features between stance categories and the ambiguity in stances, primarily describe the limitations of our approach. However, we plan to focus on the categorization of tweets, entity detection for identifying targets, and extraction of causes behind the stances in our future work to further improve the performance of the classification task. Moreover, in our current work, we do not investigate the SD task with the advancement of large language models such as ChatGPT. However, it would be interesting to investigate the performance of ChatGPT with chain-of-thought (CoT) prompting and find out if there are inherent biases in LLMs toward a particular target domain for the task SD, which would lead us in future research direction.

## Ethics Statement

Our study uses publicly available datasets and we only extend the datasets with the toxicity, morality, and speech act annotations. We obeyed the restrictions on data use and did not violate any copyright issues. Since the datasets are online-generated content, we only share the tweet IDs and annotations to comply with the terms of use and protect individual privacy.

## Acknowledgments

This work was partly funded by the SoMeCliCS project under the Volkswagen Stiftung and Niedersächsisches Ministerium für Wissenschaft und Kultur.

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

# A  Case Studies

To gain insight into the predictive performance of our approach, we present some of the predictions by different variants of our approach in Table 5. The red markers in the labels indicate incorrect predictions. Examples 1 and 3 are correctly detected by our model. The tweets do contain contrasting emotions, with the first sentence being positive but the second part containing negative emotions, leading to a hint of sarcasm in the tweet. However, the presence of dominant toxic, moral, and linguistic features found in the tweets of "Sanders against attitude" and "climate denier" correctly captures the overall intent and attitude of the tweet, thus eliminating the drawbacks of the previous works. In Example 2, it can be seen that the combination of SD +TD+ MC is not able to correctly identify the stance, as the moral features " harm" and "degradation" are more common in the "none" and "favor" stance categories of the "feminism" domain. Due to the toxic and suggestive language that is more prevalent in the "Against" and "Climate Denier" categories, SD +TD+ AC and the combinations of all tasks correctly predict attitude, justifying the higher loss weight for the tweet act than for the moral classification task. We list some possible reasons for the errors in the SD task.

## A.1  Error Scenarios

*(i) Close Proximity of auxiliary features among stances:* As shown in Examples 4 and 5, TWISTED does not correctly identify stances due to the close correspondence of attitudes with respect to the extracted virtue, vice, toxic, or act features, especially in the atheism and mask domains, which lead to scenarios in which auxiliary tasks are not helpful. The toxic, harm, subversion, and degradation features, as well as the speech act expression in Example 4, are more prevalent in the favor attitude category, while Example 5 contains

the non_toxic, harm, betrayal features that are common in the none stance category of the face_masks domain. This leads to poor performance of our TWISTED approach in these target domains compared to other basic tasks.

*(ii) Ambiguity/Conflicting Stances:* Some tweets have a contradictory attitude, either toward a single target or toward multiple targets. In Example 6, we can see that the tweeter believes that climate change is real, but does not believe in climate action by government officials. Such scenarios are misclassified by our approach, although further categorization of the target or identification of multiple targets could be further helpful for the task SD.

# B  Dataset

## B.1  Data Pre-processing

Since removing query hashtags may result in the tweet not expressing the same attitude, we first also remove the query hashtags from the datasets in accordance with previous literature (Mohammad et al., 2016). We then clean up the tweets by removing URLs, punctuation, spaces, stop words, and unwanted characters like RT and CC. To tokenize tweets, we use the NLTK-based TweetTokenizer. Tokens are converted to lowercase and then stemmed with PorterStemmer, which reduces words to their stems and suffixes.

## B.2  Percentage Distribution of features in Datasets

Tables 6, 7, 8, and 9 represent the percentage distribution of toxic (toxic/non_toxic), moral (care, harm, fairness, cheating, loyalty, betrayal, authority, subversion, purity, degradation) and speech act (expression, others, question, statement, suggestion) among the different categories of attitudes ( favor, against, none) corresponding to the SemEval-2016, P-Stance, ClimateChange, and COVID -19-Stance datasets respectively.

## B.3  Manual Verification

To examine the annotations of weakly supervised approaches, three trained annotators randomly selected 200 tweets from 4 datasets used in our study, yielding a total of 1000 tweets, and performed manual annotations to provide toxic, moral, and speech act labels (the inter-annotator agreement was calculated using the Fleiss-Kappa score and was 0.82). We then matched these manual annotations with

| No. | Tweet | True labels | Single-Task SD | SD+TD+AC | SD+TD+MC | SD+TD+ MC+AC |
|---|---|---|---|---|---|---|
| 1. | The funniest thing is these kids actually believe #BernieSanders is going to be able to pay off school debt. talks a good talk, but is literally full of sh*t (Sanders) | **stance: against** toxic moral:harm, cheating act:expression | favor | against | against | against |
| 2. | If you're a feminist it's only because absolutely no one would shove their semi hard c*** into you. Let that sink in. #SemST (Feminism) | **stance: against** toxic moral:harm, degradation act:suggestion | favor | against | favor | against |
| 3. | Glad you're are back..#Cop26 is bullshit!! the biggest #con known to man along with. (ClimateChange) | **stance: deny** toxic moral:harm, betrayal act:expression | believe | deny | deny | deny |
| 4. | I am human. I look forward to the extinction of humanity with eager anticipation. We deserve nothing less.#SemST (Atheism)" | **stance:against** toxic moral: harm, subversion, degradation act:expression | favor | favor | favor | favor |
| 5. | That's what some of us are doing Shame on you,#MasksOff (face_masks) | **stance: favor** non_toxic moral:harm,betrayal act:expression | none | none | none | none |
| 6. | Miami Crisis!! GlobalWarming is real!! PlanetDestruction Hapening!! all while the rich and famous keep buying waterfront property #BullShit #Propaganda #ClimateChangeIsBullShit (ClimateChange) | **stance: deny** toxic moral:care,betrayal act:statement | believe | believe | believe | believe |

Table 5: Sample tweets with true and predicted labels for single and multi-task models.

| Attribute | Atheism favor | Atheism against | Atheism none | Climate favor | Climate against | Climate none | Feminism favor | Feminism against | Feminism none | Hillary favor | Hillary against | Hillary none | Abortion favor | Abortion against | Abortion none |
|---|---|---|---|---|---|---|---|---|---|---|---|---|---|---|---|
| **Toxicity Detection** | | | | | | | | | | | | | | | |
| **toxic** | **31.45** | 7.33 | 26.21 | 5.68 | 7.69 | **13.31** | 44.77 | **63.99** | 29.41 | 10.19 | **27.39** | 23.78 | **32.45** | 26.19 | 27.28 |
| **non_toxic** | 68.54 | **92.67** | 73.79 | **94.32** | 92.31 | 86.69 | 55.22 | 36.01 | **70.59** | **89.81** | 72.61 | 76.22 | 67.55 | **73.81** | 72.72 |
| **Moral Classification** | | | | | | | | | | | | | | | |
| **care** | 14.52 | **16.81** | 15.86 | 14.63 | **15.38** | 12.81 | 7.84 | **8.22** | 5.88 | 13.38 | 12.76 | **18.85** | 9.27 | **10.71** | 7.18 |
| **harm** | 33.87 | **34.27** | 33.79 | 50.75 | 65.38 | 47.29 | 50.75 | 44.81 | **54.71** | 29.3 | **34.52** | 29.92 | 45.7 | 48.18 | **51.67** |
| **fairness** | 16.13 | 16.59 | **18.62** | **17.64** | 15.38 | 12.81 | 13.81 | 14.48 | **15.29** | **14.01** | 10.51 | 12.3 | **18.54** | 9.37 | 12.92 |
| **cheating** | 20.16 | 18.32 | 19.31 | 10.6 | **17.69** | 16.26 | 27.61 | **28.77** | 27.06 | 29.3 | **31.33** | 31.15 | 22.52 | **25.24** | 22.97 |
| **loyalty** | 19.35 | **22.84** | 22.76 | 14.93 | 0.0 | **17.24** | 9.7 | **12.72** | 8.82 | 21.02 | 24.02 | **24.18** | 15.23 | 13.58 | 11.48 |
| **betrayal** | 8.06 | **10.13** | 8.28 | 7.46 | 19.23 | 8.37 | 10.45 | **11.74** | 8.82 | 11.46 | 10.32 | **13.11** | 11.26 | 13.96 | **18.18** |
| **authority** | 10.48 | 13.15 | **19.31** | 9.55 | **15.38** | 12.81 | 5.6 | **9.59** | 8.82 | **14.65** | 13.88 | 12.7 | **10.6** | 8.6 | 9.57 |
| **subversion** | **11.29** | 9.7 | 6.21 | 11.94 | 0.0 | **15.27** | 13.81 | 12.33 | **15.29** | 14.01 | 14.07 | **14.75** | 9.27 | **10.71** | 5.74 |
| **purity** | **16.13** | 15.52 | 14.48 | **6.27** | 0.0 | 5.91 | 9.7 | 7.44 | **10.0** | 5.73 | 5.07 | **8.61** | 6.62 | 8.22 | **10.53** |
| **degradation** | **14.52** | 13.79 | 11.03 | 17.61 | **46.15** | 17.73 | **26.87** | 25.83 | 25.29 | 14.01 | **15.2** | 13.93 | 20.53 | 21.22 | **23.92** |
| **Speech Act Classification** | | | | | | | | | | | | | | | |
| **expression** | **33.06** | 29.31 | 27.59 | 17.01 | **26.92** | 15.27 | **35.48** | 26.22 | 21.17 | **26.11** | 18.57 | 15.57 | **48.34** | 35.76 | 21.53 |
| **others** | 19.35 | **21.34** | 18.62 | 29.55 | 19.23 | **33.49** | 17.54 | 17.61 | **31.18** | **33.12** | 31.52 | 30.74 | 12.58 | 23.52 | **29.19** |
| **question** | 4.03 | 2.59 | **4.14** | 5.37 | 0 | **7.88** | **6.34** | 6.06 | 1.76 | 3.82 | **4.88** | 3.69 | 3.97 | 4.39 | **4.78** |
| **statement** | 33.06 | 35.78 | **41.38** | 37.61 | **46.15** | 37.44 | 33.95 | **42.07** | 41.17 | 27.4 | 38.08 | **41.8** | 22.52 | 28.68 | **35.41** |
| **suggestion** | 10.48 | **10.99** | 8.28 | **10.45** | 7.69 | 5.91 | 6.72 | **8.02** | 4.07 | **9.55** | 6.94 | 8.19 | **12.59** | 7.65 | 9.09 |

Table 6: Percentage distribution of toxic, moral, and act features in SemEval-2016 dataset

| Attribute | Trump | | Biden | | Sanders | |
|---|---|---|---|---|---|---|
| | favor | against | favor | against | favor | against |
| **Toxicity Detection** | | | | | | |
| toxic | 31.94 | **49.17** | 16.66 | **42.8** | 19.18 | **48.16** |
| non_toxic | **68.06** | 50.83 | **83.34** | 57.2 | **80.82** | 51.84 |
| **Moral Classification** | | | | | | |
| care | **7.21** | 6.78 | **11.19** | 8.63 | **10.17** | 9.77 |
| harm | 43.11 | **45.67** | 39.79 | **41.6** | 38.08 | **39.1** |
| fairness | 11.66 | **11.73** | **13.49** | 13.29 | **14.31** | 12.68 |
| cheating | 32.62 | **34.46** | 28.19 | **33.0** | 30.48 | **32.14** |
| loyalty | **20.53** | 16.25 | **22.85** | 19.2 | **24.42** | 21.87 |
| betrayal | 11.44 | **12.19** | **10.66** | 10.17 | 12.11 | **12.22** |
| authority | **12.91** | 10.24 | **13.89** | 11.38 | **14.11** | 12.54 |
| subversion | 17.64 | **17.95** | 14.14 | **15.1** | **15.69** | 15.6 |
| purity | **3.82** | 3.5 | **4.63** | 4.41 | 3.49 | **3.64** |
| degradation | 14.93 | **16.09** | 15.26 | **18.46** | 11.8 | **12.97** |
| **Speech Act Classification** | | | | | | |
| expression | 24.65 | **27.07** | 55.42 | **57.83** | 40.25 | **43.56** |
| others | **25.53** | 24.32 | **2.46** | 1.08 | **15.86** | 15.17 |
| question | 0.74 | **0.86** | 0.12 | **0.27** | **4.37** | 4.36 |
| statement | **45.07** | 42.81 | **38.45** | 38.15 | **31.43** | 28.36 |
| suggestion | 4.01 | **4.93** | **3.54** | 2.67 | 8.08 | **8.54** |

Table 7: Percentage distribution of toxic, moral, and act features in P-Stance dataset

| Attribute | ClimateChange | | |
|---|---|---|---|
| | believe | deny | ambiguous |
| **Toxicity Detection** | | | |
| toxic | 3.94 | **21.65** | 5.34 |
| non_toxic | **96.06** | 78.35 | 94.66 |
| **Moral Classification** | | | |
| care | **18.67** | 14.68 | 18.15 |
| harm | 38.78 | **45.09** | 42.78 |
| fairness | **14.77** | 9.41 | 13.97 |
| cheating | **16.06** | 15.9 | 14.48 |
| loyalty | **22.38** | 20.88 | 19.44 |
| betrayal | 8.41 | **9.2** | 8.82 |
| authority | 12.9 | **14.27** | 8.76 |
| subversion | 15.6 | 15.8 | **18.44** |
| purity | **10.25** | 9.2 | 9.19 |
| degradation | 11.91 | **16.36** | 14.06 |
| **Speech Act Classification** | | | |
| expression | 44.8 | **46.45** | 37.63 |
| others | **16.3** | 14.75 | 12.6 |
| question | 3.78 | **3.83** | 3.77 |
| statement | 24.94 | 29.12 | **35.94** |
| suggestion | **10.17** | 5.84 | 10.06 |

Table 8: Percentage distribution of toxic, moral, and act features in ClimateChange dataset

| Attribute | stay_at_home | | | face_masks | | | school_closures | | | fauci | | |
|---|---|---|---|---|---|---|---|---|---|---|---|---|
| | favor | against | none | favor | against | none | favor | against | none | favor | against | none |
| **Toxicity Detection** | | | | | | | | | | | | |
| toxic | 19.47 | **34.75** | 12.79 | 40.41 | **42.07** | 34.79 | 24.88 | 14.4 | 17.23 | 42.48 | **47.21** | 40.29 |
| non_toxic | 80.53 | 62.25 | **87.21** | 59.59 | 57.93 | **65.03** | 75.12 | **85.6** | 82.77 | 57.52 | 52.79 | **59.71** |
| **Moral Classification** | | | | | | | | | | | | |
| care | **25.26** | 6.0 | 14.32 | 7.65 | 7.19 | **9.25** | **15.82** | 12.8 | 13.85 | **8.94** | 8.2 | 5.25 |
| harm | 39.47 | **59.75** | 30.31 | 47.62 | 49.55 | **49.71** | 47.48 | 39.2 | 40.62 | 50.41 | **56.89** | 51.84 |
| fairness | **20.53** | 13.75 | 13.04 | **26.55** | 20.66 | 12.43 | 28.29 | 32.8 | **34.15** | **15.65** | 10.0 | 12.6 |
| cheating | 18.95 | **31.5** | 16.37 | 20.35 | **28.59** | 26.3 | 17.72 | 17.6 | **22.77** | 26.02 | **34.75** | 30.05 |
| loyalty | **18.42** | 8.5 | 16.75 | 12.7 | 9.73 | **13.87** | 7.8 | 9.6 | **10.77** | **15.85** | 11.15 | 10.24 |
| betrayal | 3.68 | **8.75** | 8.7 | 5.48 | 6.59 | **10.69** | **21.79** | 20.8 | 16.31 | **9.96** | 9.34 | 8.79 |
| authority | 4.21 | 5.5 | **10.1** | 5.34 | 6.89 | **6.94** | 7.32 | 10.0 | **10.15** | **9.55** | 5.25 | 8.53 |
| subversion | 12.11 | **15.5** | 11.89 | 19.34 | **19.76** | 14.45 | 14.96 | 13.6 | **15.69** | **16.26** | 12.95 | 14.17 |
| purity | 17.37 | 4.75 | 7.54 | **18.04** | 11.08 | 8.67 | **7.8** | 6.4 | 4.0 | **9.76** | 4.92 | 5.51 |
| degradation | 8.95 | **19.0** | 9.85 | 12.99 | **16.92** | 16.76 | 8.46 | 6.8 | **12.92** | 17.89 | **22.95** | 22.57 |
| **Speech Act Classification** | | | | | | | | | | | | |
| expression | **44.21** | 42.75 | 25.96 | 32.61 | 31.44 | **38.43** | **44.72** | 37.2 | 42.15 | **47.97** | 41.15 | 41.47 |
| others | 6.84 | 8.5 | **37.08** | 19.19 | 18.86 | 17.63 | 8.45 | **14** | 8.31 | 11.38 | **13.28** | 11.68 |
| question | **7.36** | 7 | 7.03 | **10.53** | 9.43 | 7.51 | **8.45** | 7.6 | 7.38 | 7.32 | 6.72 | **8.01** |
| statement | **30** | 26.5 | 22.37 | 26.84 | **28.29** | 26.88 | 31.21 | 31.2 | **32.92** | 25.81 | 29.84 | **30.84** |
| suggestion | 11.58 | **15.25** | 7.54 | 10.82 | **11.97** | 9.54 | 7.16 | **10** | 9.23 | 7.52 | **9.02** | 8.01 |

Table 9: Percentage distribution of toxic, moral, and act features in COVID-19-Stance dataset

| Component | Precision | Recall | F1 score |
|---|---|---|---|
| - | Avg/St.dev | Avg/St.dev | Avg/St.dev |
| Single Task Set-up | | | |
| BERTweet | 83.19/1.61 | 86.01/2.04 | 84.72/1.58 |
| BERTweet+VAD | 87.33/0.55 | 88.21/0.69 | 87.09/1.04 |
| BERTweet+VAD +MHA | 89.51/1.37 | 87.65/0.91 | 88.55/1.15 |
| BERTweet+VAD +MHA+(toxic, moral,speech)as i/p | 90.36/1.11 | 89.17/1.42 | 90.01/1.31 |
| Multi Task Set-up | | | |
| BERTweet | 92.63/1.08 | 91.08/0.66 | 91.75/1.42 |
| BERTweet+VAD | 94.71/0.62 | 94.24/0.58 | 94.64/0.49 |
| BERTweet+VAD +MHA (TWISTED) | 95.19/0.46 | 96.11/0.23 | 96.05/0.32 |

Table 10: Significance of proposed components on ClimateChange dataset

| Component | Fauci | Home | Mask | Schools |
|---|---|---|---|---|
| Single Task Set-up | | | | |
| BERTweet | 80.07 | 79.15 | 76.82 | 75.49 |
| BERTweet+VAD | 83.45 | 82.27 | 77.81 | 79.18 |
| BERTweet+VAD +MHA | 84.73 | 83.81 | 80.05 | 81.09 |
| BERTweet+VAD +MHA+(toxic, moral,speech)as i/p | 86.01 | 84.09 | 81.64 | 82.88 |
| Multi Task Set-up | | | | |
| BERTweet | 87.14 | 86.84 | 80.67 | 81.32 |
| BERTweet+VAD | 90.06 | 89.52 | 82.11 | 85.66 |
| BERTweet+VAD +MHA (TWISTED) | 91.05 | 91.63 | 85.29 | 89.78 |

Table 11: Significance of proposed components on COVID-19 dataset

## D Significance of Components

Tables 10 and 11 justify the significance of each proposed component of our TWISTED approach. In addition, the tables 10 and 11 present the results of single-task and multi-task setups for detecting stance on the ClimateChange and COVID-19 datasets, respectively, and show the improvement in the performance of the model in addition to each component.

labels generated by semi-supervised approaches for the same 1000 tweets and found Fleiss-Kappa (Spitzer et al., 1967) scores of 0.81, 0.76, and 0.79 for toxic, moral, and speech acts, respectively, indicating that the labels predicted by semi-supervised approaches are of considerably good quality.

## C Environment Details

GPU Model: NVIDIA GeForce GTX 1080Ti GPU servers (TDP of 250W) with carbon efficiency of 0.38 $kgCO_2eq$/kWh, Library Version: Tensorflow 2.12.0, Keras 2.12.0, Transformers 4.30.2, Computational Cost: On an average, 30 minutes training time for TWISTED for one round. Average 5 rounds for each reported result for SemEval, P-Stance, and COVID, while the 5-fold cross-validation technique in Climate takes approximately 2 hours executing time, resulting in a total of 10 hours of computation, leading to $\approx 0.95$ $kgCO_2eq$ carbon emissions. Carbon footprint is calculated using the Machine Learning Impact calculator (Lacoste et al., 2019).