# OpenReview forum: "Toxicity, Morality, and Speech Act Guided Stance Detection"
_EMNLP/2023/Conference — EMNLP 2023 Findings_

### Official Review · Reviewer_ygJA · 2023-07-31

**Soundness:** 2

**Excitement:**

2: Mediocre: This paper makes marginal contributions (vs non-contemporaneous work), so I would rather not see it in the conference.

**Missing References:**

Islam et al 2023 (Analysis of Climate Campaigns on Social Media using Bayesian Model Averaging).

Islam et al 2023 (Weakly Supervised Learning for Analyzing Political Campaigns on Facebook).

Pacheco et al 2022 (A holistic framework for analyzing the covid-19 vaccine debate).

Islam and Goldwasser 2022 (Understanding COVID-19 Vaccine Campaign on Facebook using Minimal Supervision).

**Paper Topic And Main Contributions:**

The authors presented a stance detection task using the associated toxic, moral, and speech act traits present in tweets (they proposed a multitasking model named TWISTED). They conducted  4 benchmark datasets to show that multi-tasking enhances the performance of the stance task compared to the single-task and baseline methods by leveraging the auxiliary tasks.

**Reasons To Accept:**

Interesting use of toxicity, morality, and speech acts presented in the tweet as auxiliary tasks of detecting stance (main task).

**Reasons To Reject:**

1. Small-scale contribution. This work is not much different than MEMOCLiC (recent work named "A multi-task model for emotion and offensive aided stance detection of climate change tweets."). They just injected the VAD features.
2. Lack of qualitative analysis.
3. Need more exploration in related work. Authors should explore more works related to stance and moral foundation especially on climate change, politics and COVID-19 related work.

Authors should include the stance detection task on climate change by Islam et al 2023 (Analysis of Climate Campaigns on Social Media using Bayesian Model Averaging). In this work, authors showed how theme information improves stance detection.

Authors should explore the stance detection task on political campaigns by Islam et al 2023 (Weakly Supervised Learning for Analyzing Political Campaigns on Facebook).

Authors should include the stance detection task on COVID-19 by Pacheco et al 2022 (A holistic framework for analyzing the covid-19 vaccine debate). In this work authors have annotation both for stance and moral foundation as well as morality.

Also for COVID-19 moral foundation, authors could look into work by Islam and Goldwasser 2022 (Understanding COVID-19 Vaccine Campaign on Facebook using Minimal Supervision).

**Reproducibility:**

3: Could reproduce the results with some difficulty. The settings of parameters are underspecified or subjectively determined; the training/evaluation data are not widely available.

**Reviewer Confidence:**

4: Quite sure. I tried to check the important points carefully. It's unlikely, though conceivable, that I missed something that should affect my ratings.

---

> ### Author Rebuttal · Authors · 2023-08-28
>
> **1.** We have already compared the performance of the MEMOCliC model proposed in the recent work “A multi-task model for emotion and offensive aided stance detection of climate change tweets” on SemEval-2016 and ClimateChange datasets which authors of MEMOCLiC used to evaluate their model performance. The experimental results show that our model outperforms the MEMOCLiC approach. The model has been proposed for the climate change domain and we have clearly mentioned and compared our approach with the MEMOCliC pointing out its shortcomings as well as benefits from our approach in 2 sections of the manuscript:
>
>    * **Section 1 Introduction:**
>    *Other recent works such as MEMOCLiC \cite{upadhyaya2023multi}, SP-AMT \cite{upadhyaya2023multisenti} have used……valence,
>     arousal, and dominance features help in analyzing emotional tone and its intensity…. supports the detection of hidden intent or any
>      sarcasm………thus addressing the drawbacks of the presence of sarcasm and implicit stance from previous works.*
>    * **Section 5 ClimateChange:**
>     *Although MEMOCLiC extracts the emotional and toxic features….., but the performance power of our EmoSense injector, which injects
>     the VAD features, understanding ….hidden moral and linguistic features, is one of the main reasons contributing to the success of
>     TWISTED compared to the recent works MEMOCLiC, SP-AMT.*
>
> **2.** We have incorporated the qualitative analysis of our model by mentioning the case studies in Appendix A “Case Studies”, discussing some of the predictions of the different variants of our proposed approach. We have already presented the ablation study under the heading “Ablation Study” in Section 5, where we examine and present the F1 values (Figure 5) of the single task, all auxiliary tasks, and their combinations for the stance task. We also plan to include the ablation study, which shows the importance of proposing the components for our TWISTED model and justifies the motivation behind the use of the BERTweet, VAD, and MultiHeadAttention components (as highlighted by Reviewer 9nz9). Due to time constraints during the rebuttal period, we currently implemented the results for the ClimateChange dataset (refer to Table 1 mentioned in Reviewer 9nz9 rebuttal). However, we will implement the results for all datasets and include them in the manuscript.
>
>
>
> **3.** We thank the reviewer for suggesting the references. As indicated in the comment, one of the papers (A Holistic Framework for Analyzing the COVID-19 Vaccine Debate) examines the stance, reason, and moral grounds in COVID tweets, while another (Understanding COVID-19 Vaccine Campaign on Facebook using Minimal Supervision) proposes a multitasking setup for identifying themes and moral foundations for Facebook ads related to COVID. The works are quite interesting and inspire us to look into the direction of how ads and themes are relevant to the formation of public opinion as part of future work. Since our main focus in the current work is to identify the attitude of the public towards different target areas, we use the interdependencies between moral foundations, toxicity, and speech acts as auxiliary tasks for the main stance task. We have also tried to incorporate relevant and popular works for all datasets and domains we use in our manuscript. However, we will certainly include the suggested references in our manuscript and explore the dimensions mentioned in these works as part of our future research direction.

---

### Official Review · Reviewer_9nz9 · 2023-08-01

**Typos Grammar Style And Presentation Improvements:** None
**Soundness:** 4

**Excitement:**

4: Strong: This paper deepens the understanding of some phenomenon or lowers the barriers to an existing research direction.

**Missing References:**

None

**Paper Topic And Main Contributions:**

The main contribution of this paper lies in the development and evaluation of a novel approach to stance detection, known as TWISTED, which leverages additional tasks such as toxicity, morality, and speech act to enhance stance detection performance. The authors propose a multi-task learning setup where these auxiliary tasks inform the primary task of stance detection.

This approach is applied to data from Twitter; the proposed model is evaluated on four different datasets, each targeting different topics, and the results show that it outperforms several baseline models, demonstrating the effectiveness of this approach.

**Questions For The Authors:**

The one question I would have, less for a rebuttal but more for the final version of the paper, is providing additional motivation for each of the added layers. Can you provide any more intuition on why you set the model up as you did?

**Reasons To Accept:**

+ Sound and clear methods - the approach is straightfoward (in a good way) and intuitive (also in a good way), and well explained

+ Reasonable evaluation - the evaluation is comprehensive in both data and comparison methods

+ Results show considerable improvement over baselines



**Reasons To Reject:**

- There's nothing groundbreaking here. I don't think this is necessarily a significant reason to reject, but worth noting that this paper feels somewhat incremental in terms of the conceptual novelty of leveraging affect and a multi-task setup.  Nonetheless, teh contribution *is* novel

**Reproducibility:**

5: Could easily reproduce the results.

**Reviewer Confidence:**

3: Pretty sure, but there's a chance I missed something. Although I have a good feel for this area in general, I did not carefully check the paper's details, e.g., the math, experimental design, or novelty.

---

> ### Author Rebuttal · Authors · 2023-08-28
>
> We thank the reviewer for complementing our work on the grounds of approach, intuition, evaluation, and performance improvement.
>
> As pointed out by the reviewer **“Can you provide any more intuition on why you set the model up as you did?”**, We plan to add an ablation study that justifies the significance of the components used in the proposed model (BERTweet, BERTweet+VAD, BERTweet+VAD+MHA in both single-task and multi-task set-up). We currently implement the results for the ClimateChange dataset due to time constraints during the rebuttal period as it does not contain any sub-domains like other datasets, however, we will implement the results for all datasets and include them in the manuscript. Table 1 proves the importance of each component and shows the improvement in the model’s performance in addition to each component on the ClimateChange dataset.
>
> | Component  | Precision | Recall | F1 Score |
> |---------|----------------|--------|----------------|
> |---------|(Avg./Std. dev)|(Avg./Std. dev)|(Avg./Std. dev)|
> |**Single-Task Setup**|
> |BERTweet|83.19/1.61|86.01/2.04|	84.72/1.58|
> |BERTweet+VAD|87.33/0.55|88.21/0.69|87.09/1.04|
> |BERTweet+VAD+MHA|89.51/1.37|87.65/0.91|88.55/1.15|
> |BERTweet+VAD+MHA+[toxic,moral,speech act] as input features|90.36/1.11|89.17/1.42|90.01/1.31|
> |**Multi-Task Setup**|
> |BERTweet|92.63/1.08|91.08/0.66|	91.75/1.42|
> |BERTweet+VAD|94.71/0.62|94.24/0.58|94.64/0.49|
> |BERTweet+VAD+MHA (Ours TWISTED)|95.19/0.46|96.11/0.23|96.05/0.32|
> Table 1:  Results of Single-Task and Multi-Task Stance Detection on ClimateChange dataset in various combinations to show the significance of each component used in our TWISTED approach
>
> Since, we use Twitter data, the BERTweet encoder specially trained on tweets, helps encode the input text in a better way as mentioned in previous literature. We investigate that the VAD features reflect the emotional sense of the tweet. In this manner, we capture the emotional meaning of the input tweet by using these features in conjunction with the text of the tweet. Instead of simply averaging the encoded text and VAD features, which might lose the influence of valence, arousal, and dominance features over the text, we use the Emo-Sense Injector module, which acts as an integrating module and fuses the text and VAD features using the absolute difference and element-wise product to capture the entire emotional essence of the tweet in terms of the semantic and syntactic representation of the tweet text and not miss any related features by simply averaging or concatenating functions. Then, we use the MultiHeadAttention layer rather than the single attention layer to focus on all of the relevant parts of the emotional-sense injected tweet from all subspaces so that we do not avoid any important feature. While we have tried to mention all these details in Section 3 (Methodology), we will elaborate on these details in all subsections of Section 3 to make the motivation even clearer. We also refer to the tables for all datasets similar to those presented above (Table 1) under the heading “Significance of the proposed components”.
>
> We also thank the reviewer for pointing out that our contribution is novel and agree that our manuscript leverages the multi-task setup. To the best of our knowledge, this is the first cross-sectional study that exploits toxic, moral, and speech act aspects hidden within the tweet as auxiliary tasks and fuses the VAD features followed by the efficient encoding and attention frameworks to perform the main task of stance detection on various datasets comprising of different domains and targets.

---

### Official Review · Reviewer_o8Ty · 2023-08-07

**Soundness:** 3

**Excitement:**

3: Ambivalent: It has merits (e.g., it reports state-of-the-art results, the idea is nice), but there are key weaknesses (e.g., it describes incremental work), and it can significantly benefit from another round of revision. However, I won't object to accepting it if my co-reviewers champion it.

**Missing References:**

https://aaai.org/ojs/index.php/ICWSM/article/view/7334
https://arxiv.org/pdf/1909.09758.pdf
https://aclanthology.org/D19-1657.pdf
https://link.springer.com/article/10.1007/s42979-021-00455-5


**Paper Topic And Main Contributions:**

The authors propose using word count features based on Valance, Arousal, and Dominace alongside a BERT Encoder and a multi-task objective based on Toxicity, Morality, and Speech Act to demonstrate significant performance improvement on Stance Detection tasks for Tweets.

**Reasons To Accept:**

Well written paper
Good extensive experiments
Significant improvements over other stated methods
Inclusion of qualitative examples around improvements using the various features in appendix


**Reasons To Reject:**

Including Neural + Lexical features is quite common in NLP
MTL is not a novel technique and has been shown to work on other tasks including multiple tweet specific tasks. See missing references section

**Reproducibility:**

3: Could reproduce the results with some difficulty. The settings of parameters are underspecified or subjectively determined; the training/evaluation data are not widely available.

**Reviewer Confidence:**

5: Positive that my evaluation is correct. I read the paper very carefully and I am very familiar with related work.

---

> ### Author Rebuttal · Authors · 2023-08-28
>
> We thank the reviewer for the comment. We agree with the reviewer that multi-task learning is not a novel technique and has been utilized in previous works. However, the use of toxicity detection together with morality and speech act classification tasks for the main task of stance detection has not yet been explored in the existing literature. In our work, we focus on showing the importance of using toxic, moral, and speech act aspects associated with the tweet and utilize these interdependencies to understand the overall intent of the tweet, which ultimately captures the attitude of the tweet efficiently. Moreover, the valence, arousal, and dominance features that give emotional meaning to the tweet are leveraged by the efficient embedding technique and attention framework, which also improves the performance of our proposed approach in classifying the attitude of the input tweet.
>
> **Missing References**: We plan to cite the three mentioned works in the missing references: “Empirical Analysis of Multi-Task Learning for Reducing Identity Bias in Toxic Comment Detection” (work is mentioned twice with both arxiv and published versions), “Multi-Task Stance Detection with Sentiment and Stance Lexicons”, “Exploring Multi-Task Multi-Lingual Learning of Transformer Models for Hate Speech and Offensive Speech Identification in Social Media” that show the efficacy of using multi-task architecture for various other tasks. We would like to notify that 2 of the missing references are not directly related to the main task of stance and explore the multi-task set-up, where we tried to refer to works that use multi-task architecture and proposed their models for the stance task as baselines such as MT-LRM-BERT, SP-AMT, STASY, MEMOCLiC, AT-JSS-LEX, and S-MDMT methods. We also have used the AT-JSS-LEX model proposed in the “Multi-Task Stance Detection with Sentiment and Stance Lexicons” work as a baseline method that directly correlates with the stance task for the SemEval dataset and our model outperforms the AT-JSS-LEX model. Further, we did not mention AT-JSS-LEX results on the Climate Change dataset as other recent models such as SP-AMT (ICWSM-2023) and MEMOCLiC (WWW/ACM Web 2023) mentioned in our paper have already outperformed the AT-JSS-LEX model, and we used these SP-AMT and MEMOCLiC models in our work as baseline methods. However, we will surely mention all the works in the related section.
>
> As mentioned by the **“Excitement”** rating of 3 that work can benefit from another round of revision, we plan to include the ablation study that shows the significance of proposing the components for our TWISTED model and justifies the intuition and motivation behind using BERTweet, VAD, and MultiHeadAttention frameworks in both single and multi-task setup (as pointed out by the Reviewer 9nz9). We currently implement the results for the ClimateChange dataset due to time constraints during the rebuttal period (refer to Table 1 mentioned in Reviewer 9nz9 rebuttal), however, we will implement the results for other datasets as well and include them in the appropriate appendix/Results section depending on the space constraints.
>
> Regarding **reproducibility**, we already share the dataset (with annotations) and code URL in the submitted manuscript for the reviewers, and will make the code publicly available so as to reproduce the results. We mention precisely all the hyperparameters required to implement our model in Section 4.2. In line with the existing works, we use already available train, validation, and test splits of the SemEval-2016, P-Stance, and COVID-19-Stance datasets and implement our approach with 5-fold cross-validation technique for the ClimateChange dataset (these details are mentioned in Section 4.2 under the heading “Evaluation Metrics”). For the users still facing any issues in reproducibility, we intend to make the pre-trained model available to those users so that anyone who tries to reproduce the results can easily use the pre-trained model without any issues.

---

### Meta-Review · Area_Chair_mQ9K · 2023-09-18

**Recommendation:** 2

**Metareview:**

This paper proposes a new approach to social media stance detection based on speech act, toxicity, and moral features of tweets. In extensive experiments, they demonstrate that multi-task learning improves stance detection.

The following concerns have been raised by reviewers:
* It does not seem that the proposed methods are truly novel. This is because lexical and neural features and MTL are quite common in NLP.
 * Lack of qualitative analysis and small-scale experiment
 * References and comparisons with prior works are missing.
    - Empirical Analysis of Multi-Task Learning for Reducing Identity Bias in Toxic Comment Detection
    - A multi-task model for emotion and offensive aided stance detection of climate change tweets.
    - Exploring Multi-Task Multi-Lingual Learning of Transformer Models for Hate Speech and Offensive Speech Identification in Social Media
    - Task Stance Detection with Sentiment and Stance Lexicons

---

### Decision · Program_Chairs · 2023-10-07

**Decision:**

Accept-Findings

**Comment:**

This paper proposes a new approach to social media stance detection based on speech act, toxicity, and moral features of tweets. In extensive experiments, they demonstrate that multi-task learning improves stance detection.

The following concerns have been raised by reviewers:
* It does not seem that the proposed methods are truly novel. This is because lexical and neural features and MTL are quite common in NLP.
 * Lack of qualitative analysis and small-scale experiment
 * References and comparisons with prior works are missing.
    - Empirical Analysis of Multi-Task Learning for Reducing Identity Bias in Toxic Comment Detection
    - A multi-task model for emotion and offensive aided stance detection of climate change tweets.
    - Exploring Multi-Task Multi-Lingual Learning of Transformer Models for Hate Speech and Offensive Speech Identification in Social Media
    - Task Stance Detection with Sentiment and Stance Lexicons